# What is Flagged in Uncertainty Quantification? Latent Density Models for Uncertainty Categorization

**Hao Sun**[†,*] **Boris van Breugel**[†]**, Jonathan Crabbé, Nabeel Seedat, Mihaela van der Schaar**
Department of Applied Mathematics and Theoretical Physics
University of Cambridge

## Abstract

Uncertainty Quantification (UQ) is essential for creating trustworthy machine learning models. Recent years have seen a steep rise in UQ methods that can flag suspicious examples, however, it is often unclear what exactly these methods identify. In this work, we propose a framework for categorizing uncertain examples flagged by UQ methods in classification tasks. We introduce the confusion density matrix—a kernel-based approximation of the misclassification density—and use this to categorize suspicious examples identified by a given uncertainty method into three classes: out-of-distribution (OOD) examples, boundary (Bnd) examples, and examples in regions of high in-distribution misclassification (IDM). Through extensive experiments, we show that our framework provides a new and distinct perspective for assessing differences between uncertainty quantification methods, thereby forming a valuable assessment benchmark.

## 1 Introduction

Black-box parametric models like neural networks have achieved remarkably good performance on a variety of challenging tasks, yet many real-world applications with safety concerns—e.g. healthcare [1], finance [2, 3] and autonomous driving [4, 5]—necessitate reliability. These scenarios require trustworthy model predictions to avoid the high cost of erroneous decisions.

Uncertainty quantification [6–10] addresses the challenge of trustworthy prediction through inspecting the confidence of a model, enabling intervention whenever uncertainty is too high. Usually, however, the cause of the uncertainty is not clear. In this work, we go beyond black-box UQ in the context classification tasks and aim to answer two questions:

1. How do we provide a more granular categorization of *why* uncertainty methods to identify certain predictions as suspicious?
2. *What* kind of examples do different UQ methods tend to mark as suspicious?

**Categorizing Uncertainty**   We propose a Density-based Approach for Uncertainty Categorization (DAUC): a model-agnostic framework that provides post-hoc categorization for model uncertainty. We introduce the confusion density matrix, which captures the predictive behaviors of a given model. Based on such a confusion density matrix, we categorize the model's uncertainty into three classes: (1) **OOD** uncertainty caused by OOD examples, i.e. test-time examples that resemble no training-time sample [11–14]. Such uncertainty can be manifested by a low density in the confusion density matrix; (2) **Bnd** uncertainty caused by neighboring decision boundaries, i.e., non-conformal predictions due to confusing training-time resemblances from different classes or inherent ambiguities making it challenging to classify the data [15–17]; Such uncertainty can be manifested by the density of

---

*hs789@cam.ac.uk, †: equal contributions

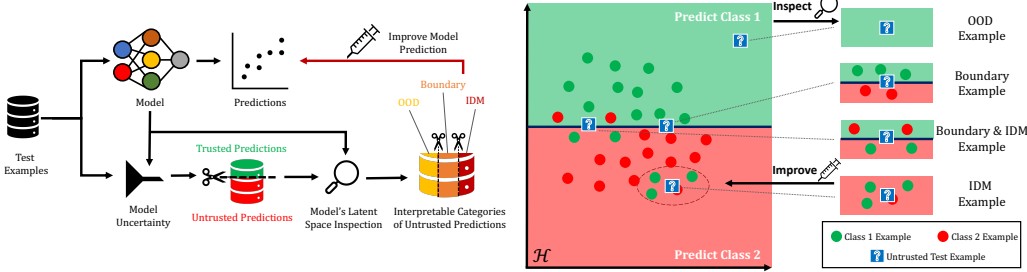

Figure 1: Given a prediction and UQ algorithm, our method divides flagged test time examples into different classes. Class **OOD** identifies outliers, which are mistrusted because they do not resemble training data; class **IDM** indicates examples lying in regions with high misclassification; class **Bnd** indicates examples that lie near the decision boundary.

diagonal elements in the confusion density matrix; (3) **IDM** uncertainty caused by imperfections in the model—as manifested by high misclassification at validation time—such that similar misclassification is to be expected during testing [18]. Such uncertainty can be manifested by the density of off-diagonal elements in the confusion density matrix. Figure 1 illustrates the different classes and in Section 4 we provide visualisations using real data. We show how DAUC can be used to benchmark a broad class of UQ methods, by categorizing what each method tends to flag as suspicious.

**Our contributions**   can be summarized as follows:

1. Formally, we propose the confusion density matrix—the heart of DAUC—that links the training time error, decision boundary ambiguity, and uncertainty with latent representation density.
2. Practically, we leverage DAUC as a unified framework for uncertain example categorization. DAUC offers characterisation of uncertain examples at test time.
3. Empirically, we use DAUC to benchmark existing UQ methods. We manifest different methods' sensitivity to different types of uncertain examples, this provides model insight and aids UQ method selection.

## 2   Related Work

**Uncertainty Quantification**   Uncertainty quantification methods are used to assess the confidence in a model's predictions. In recent years the machine learning community has proposed many methods which broadly fall into the following categories: (1) Ensemble methods (i.e. Deep Ensembles [6]), which—while considered state of the art—have a high computational burden; (2) Approximate Bayesian methods (i.e. Stochastic Variational Inference [7–9]), however work by [19–21] suggests these methods do not yield high-quality uncertainty estimates; and (3) Dropout-based methods such as Monte Carlo Dropout [10], which are simpler as they do rely on estimating a posterior, however the quality of the uncertainty estimates is linked to the choice of parameters which need to be calibrated to match the level of uncertainty and avoid suboptimal performance [22]. For completeness we note that whilst conformal prediction [23] is another UQ method, the paradigm is different from the other aforementioned methods, as it returns predictive sets to satisfy coverage guarantees rather than a value-based measure of uncertainty.

In practice, the predictive uncertainty for a model prediction arises from the lack of relevant training data (epistemic uncertainty) or the inherent non-separable property of the data distribution (aleatoric uncertainty) [24–26]. This distinction in types of uncertainty is crucial as it has been shown that samples with low epistemic uncertainty are more likely under the data distribution, hence motivating why epistemic uncertainty has been used for OOD/outlier detection [10]. Moreover, ambiguous instances close to the decision boundary typically result in high aleatoric uncertainty [27].

This suggests that different sets of uncertain data points are associated with different types of uncertainties and, consequently, different types of misclassifications. Furthermore, it is to be expected that different uncertainty estimators are more able/prone to capturing some types of uncertainty than others. To understand these models better, we thus require a more granular definition to characterize

Table 1: Comparison with related work in UQ. We aim to provide a flexible framework for inspecting mistrusted examples identified by uncertainty estimators. Furthermore, this framework enables us to improves the prediction performance on a certain type of flagged uncertain class.

| Method | Model Structure | Uncertainty Estimation | Categorize Uncertainty | Improve Prediction | Examples |
|---|---|---|---|---|---|
| BNNs | Bayesian Layers | ✓ | · | · | [8, 9] |
| GP | Gaussian Processes | ✓ | · | · | [39] |
| MC-Dropout | Drop-Out Layers | ✓ | · | · | [10] |
| Deep-Ensemble | Multiple Models | ✓ | · | · | [6] |
| ICP | Model "Wrapper" | ✓ | · | · | [23, 40], |
| Performance Prediction | Multiple Predictive Models | ✓ | · | · | [37, 18] |
| DAUC | Assumption 1 | ✓ | ✓ | ✓ | **(Ours)** |

uncertain data points. We employ three classes: outliers (OOD), boundary examples (Bnd) and examples of regions with high in-distribution misclassification (IDM), see Fig 1.

**Out-of-distribution (OOD) detection**   Recall that UQ methods have been used to flag OOD examples. For completeness, we highlight that other alternative methods exist for OOD detection. For example, Lee et al. [28] detects OOD examples based on the Mahalanobis distance, whilst Ren et al. [29] uses the likelihood ratio between two generative models. Besides supervised learning, OOD is an essential topic in offline reinforcement learning [30–35]. Sun et al. [36] detects the OOD state in the context of RL using confidence intervals. We emphasize that DAUC's aim is broader—creating a unifying framework for categorizing multiple types of uncertainty—however, existing OOD methods could be used to replace DAUC's OOD detector.

**Accuracy without Labels**   We contrast our work to the literature which aims to determine model accuracy without access to ground-truth labels. Methods such as [37, 38] propose a secondary regression model as an accuracy predictor given a data sample. Ramalho and Miranda [18] combine regression model with latent nearest neighbors for uncertain prediction. This is different from our setting which is focused on inspecting UQ methods on a sample level by characterizing it as an outlier, boundary, or IDM example. We contrast DAUC with related works in Table 1.

## 3   Categorizing Model Uncertainty via Latent Density

### 3.1   Preliminaries

We consider a typical classification setting where $\mathcal{X} \subseteq \mathbb{R}^{d_X}$ is the input space and $\mathcal{Y} = [0, 1]^C$ is the set of class probabilities, where $d_X$ is the dimension of input space and $C \in \mathbb{N}^*$ is the number of classes. We are given a prediction model $f : \mathcal{X} \to \mathcal{Y}$ that maps $x \in \mathcal{X}$ to class probabilities $f(x) \in \mathcal{Y}$. An uncertainty estimator $u : \mathcal{X} \to [0, 1]$ quantifies the uncertainty of the predicted outcomes. Given some threshold $\tau$, the inference-time predictions can be separated into trusted predictions $\{x \in \mathcal{X} | u(x) < \tau\}$ and untrusted predictions $\{x \in \mathcal{X} | u(x) \geq \tau\}$. We make the following assumption on the model architecture of $f$.

**Assumption 1** (Model Architecture). *Model $f$ can be decomposed as $f = \varphi \circ l \circ g$, where $g : \mathcal{X} \to \mathcal{H} \subseteq \mathbb{R}^{d_H}$ is a feature extractor that maps the input space to a $d_H < d_X$ dimensional latent (or representation) space, $l : \mathcal{H} \to \mathbb{R}^C$ is a linear map between the latent space to the output space and $\varphi : \mathbb{R}^C \to \mathcal{Y}$ is a normalizing map that converts vectors into probabilities.*

**Remark 1.** *This assumption guarantees that the model is endowed with a lower-dimensional representation space. Most modern uncertainty estimation methods like MCD, Deep-Ensemble, BNN satisfy this assumption. In the following, we use such a space to categorize model uncertainty.*

We assume that the model and the uncertainty estimator have been trained with a set of $N \in \mathbb{N}^*$ training examples $\mathcal{D}_{\text{train}} = \{(x^n, y^n) \mid n \in [N]\}$. At inference time the underlying model $f$ and uncertainty method $u$ predict class probabilities $f(x) \in \mathcal{Y}$ and uncertainty $u(x) \in [0, 1]$, respectively. We assign a class to this probability vector $f(x)$ with the map $\text{class} : \mathcal{Y} \to [C]$ that maps a probability vector $y$ to the class with maximal probability $\text{class}[y] = \arg\max_{c \in [C]} y_c$. While uncertainty estimators flag examples to be trustworthy or not, those estimators do not provide a fine-grained reason for what a certain prediction should not be mistrusted. Our aim is to use the

model's predictions and representations of a corpus of labelled examples—which we will usually take to be the training ($\mathcal{D}_{\text{train}}$) or validation ($\mathcal{D}_{\text{val}}$) sets—to categorize inference-time uncertainty predictions. To that aim, we distinguish two general scenarios where a model's predictions should be considered with skepticism.

## 3.2 Flagging OOD Examples

There is a limit to model generalization. Uncertainty estimators should be skeptical when the input $x \in \mathcal{X}$ differs significantly from input examples that the model was trained on. From an UQ perspective, the predictions for these examples are expected to be associated with a large epistemic uncertainty.

A natural approach to flagging these examples is to define a density $p\left(\cdot \mid \mathcal{D}_{\text{train}}\right) : \mathcal{X} \to \mathbb{R}^+$ over the input space. This density should be such that $p\left(x \mid \mathcal{D}_{\text{train}}\right)$ is high whenever the example $x \in \mathcal{X}$ resembles one or several examples from the training set $\mathcal{D}_{\text{train}}$. Conversely, a low value for $p\left(x \mid \mathcal{D}_{\text{train}}\right)$ indicates that the example $x$ differs from the training examples. Of course, estimating the density $p\left(x \mid \mathcal{D}_{\text{train}}\right)$ is a nontrivial task. At this stage, it is worth noting that this density does not need to reflect the ground-truth data generating process underlying the training set $\mathcal{D}_{\text{train}}$. For the problem at hand, this density $p\left(x \mid \mathcal{D}_{\text{train}}\right)$ need only measure how close the example $x$ is to the training data manifold. A common approach is to build a kernel density estimation with the training set $\mathcal{D}_{\text{train}}$. Further, we note that Assumption 1 provides a representation space $\mathcal{H}$ that was specifically learned for the classification task on $\mathcal{D}_{\text{train}}$. In Appendix A.1, we argue that this latent space is suitable for our kernel density estimation. This motivates the following definition for $p\left(\cdot \mid \mathcal{D}_{\text{train}}\right)$.

**Definition 1** (Latent Density). *Let $f : \mathcal{X} \to \mathcal{Y}$ be a prediction model, let $g : \mathcal{X} \to \mathcal{H}$ be the feature extractor from Assumption 1 and let $\kappa : \mathcal{H} \times \mathcal{H} \to \mathbb{R}^+$ be a kernel function. The* latent density *$p\left(\cdot \mid \mathcal{D}\right) : \mathcal{X} \to \mathbb{R}^+$ is defined over a dataset $\mathcal{D}$ as:*

$$p\left(x \mid \mathcal{D}\right) \equiv \frac{1}{N} \sum_{\tilde{x} \in \mathcal{D}} \kappa\left[g(x), g(\tilde{x})\right] \tag{1}$$

Test examples with low training density are likely to be underfitted for the model — thus should not be trusted.

**Definition 2** (OOD Score). *The* OOD Score *$T_{\text{OOD}}$ is defined as*

$$T_{\text{OOD}}(x) \equiv \frac{1}{p(x|\mathcal{D}_{\text{train}})} \tag{2}$$

For a test example $x \in \mathcal{D}_{\text{test}}$, if $T_{\text{OOD}}(x|\mathcal{D}_{\text{train}}) \geq \tau_{\text{OOD}}$ the example is suspected to be an outlier with respect to the training set. We set $\tau_{\text{OOD}} = \frac{1}{\min_{x' \in \mathcal{D}_{\text{train}}} p(x'|\mathcal{D}_{\text{train}})}$, i.e. a new sample's training density is smaller than the minimal density of training examples.

## 3.3 Flagging IDM and Boundary Examples

Samples that are not considered outliers, yet are given high uncertainty scores, we divide up further into two non-exclusionary categories. The first category consists of points located near the boundary between two or more classes, the second consists of points that are located in regions of high misclassification.

For achieving this categorization, we will use a separate validation set $\mathcal{D}_{\text{val}}$. We first partition the validation examples according to their true and predicted label. More precisely, for each couple of classes $(c_1, c_2) \in [C]^2$, we define the corpus $\mathcal{C}_{c_1 \mapsto c_2} \equiv \{(x, y) \in \mathcal{D}_{\text{val}} \mid \text{class}[y] = c_1 \wedge \text{class}[f(x)] = c_2\}$ of validation examples whose true class is $c_1$ and whose predicted class is $c_2$. In the case where these two classes are different $c_1 \neq c_2$, this corresponds to a corpus of misclassified examples. Clearly, if some example $x \in \mathcal{X}$ resembles one or several examples of those misclassification corpus, it is legitimate to be skeptical about the prediction $f(x)$ for this example. In fact, keeping track of the various misclassification corpora from the validation set allows us to have an idea of what the misclassification is likely to be.

In order to make this detection of suspicious examples quantitative, we will mirror the approach from Definition 1. Indeed, we can define a kernel density $p\left(\cdot \mid \mathcal{C}_{c_1 \mapsto c_2}\right) : \mathcal{X} \to \mathbb{R}^+$ for each corpus

$\mathcal{C}_{c_1 \mapsto c_2}$. Again, this density will be such that $p\left(\cdot \mid \mathcal{C}_{c_1 \mapsto c_2}\right)$ is high whenever the representation of the example $x \in \mathcal{X}$ resembles the representation of one or several examples from the corpus $\mathcal{C}_{c_1 \mapsto c_2}$. If this corpus is a corpus of misclassified examples, this should trigger our skepticism about the model's prediction $f(x)$. By aggregating the densities associated with each of these corpora, we arrive at the following definition.

**Definition 3** (Confusion Density Matrix). *Let $f : \mathcal{X} \to \mathcal{Y}$ be a prediction model, let $g : \mathcal{X} \to \mathcal{H}$ be the feature extractor from Assumption 1 and let $\kappa : \mathcal{H} \times \mathcal{H} \to \mathbb{R}^+$ be a kernel function. The confusion density matrix $P(\cdot \mid \mathcal{D}_{\mathrm{val}}) : \mathcal{X} \to (\mathbb{R}^+)^{C \times C}$ is defined as*

$$P_{c_1, c_2}\left(x \mid \mathcal{D}_{\mathrm{val}}\right) \equiv p\left(x \mid \mathcal{C}_{c_1 \mapsto c_2}\right) = \frac{1}{|\mathcal{C}_{c_1 \mapsto c_2}|} \sum_{\tilde{x} \in \mathcal{C}_{c_1 \mapsto c_2}} \kappa[g(x), g(\tilde{x})], \forall (c_1, c_2) \in [C]^2 \quad (3)$$

**Remark 2.** *The name* confusion density *is chosen to make a parallel with confusion matrices. Like confusion matrices, our confusion density indicates the likelihood of each couple $(c_1, c_2)$, where $c_1$ is the true class and $c_2$ is the predicted class. Unlike confusion matrices, our confusion density provides an instance-wise (i.e. for each $x \in \mathcal{X}$) likelihood for each couple.*

### 3.3.1 Bnd examples

By inspecting the confusion matrix, we can quantitatively distinguish two situations where the example $x \in \mathcal{X}$ is likely to be mistrusted. The first situation where high uncertainty arises, is when $g(x)$ is close to latent representations of validation examples that have been correctly assigned a label that differs from the predicted one $\mathrm{class}[f(x)]$. This typically happens when $g(x)$ is located close to a decision boundary in latent space. In our validation confusion matrix, this likelihood that $x$ is related to validation examples with different labels is reflected by the diagonal elements. This motivates the following definition.

**Definition 4** (Boundary Score). *Let $P(x \mid \mathcal{D}_{\mathrm{val}})$ be the* confusion density matrix *for an example $x \in \mathcal{X}$ with predicted class $\hat{c} = \mathrm{class}[f(x)]$. We define the* boundary score *as the sum of each density of well-classified examples from a different class:*

$$T_{\mathrm{Bnd}}(x) = \sum_{c \neq \hat{c}}^{C} P_{c,c}\left(x \mid \mathcal{D}_{\mathrm{val}}\right). \quad (4)$$

Points are identified as Bnd when $T_{\mathrm{Bnd}} > \tau_{\mathrm{Bnd}}$—see Appendix A.2.

### 3.3.2 IDM examples

The second situation is the one previously mentioned: the latent representation $g(x)$ is close to latent representations of validation examples that have been misclassified. In our validation confusion matrix, this likelihood that $x$ is related to misclassified validation examples is reflected by the off-diagonal elements. This motivates the following definition.

**Definition 5** (IDM Score). *Let $P(x \mid \mathcal{D}_{\mathrm{val}})$ be the* confusion density matrix *for an example $x \in \mathcal{X}$. We define the* IDM score *as the sum of each density corresponding to a misclassification of the predicted class $\hat{c} = \mathrm{class}\, f(x)$ in the confusion density matrix:*

$$T_{\mathrm{IDM}}(x) = \sum_{c \neq \hat{c}}^{C} P_{c,\hat{c}}\left(x \mid \mathcal{D}_{\mathrm{val}}\right). \quad (5)$$

Points are identified as IDM when $T_{\mathrm{IDM}} > \tau_{\mathrm{IDM}}$. We choose $\tau_{\mathrm{IDM}}$ such that the proportion of IDM points in the validation set equals the number of misclassified examples. Details for definitions and choices of thresholds are provided in Appendix A.2.

**Remark 3.** *Note that the definitions of Bnd examples and IDM examples do not exclude each other, therefore, an uncertain example can be flagged as a Bnd example, an IDM example, or flagged as both Bnd and IDM (B&I). To make this distinction clear, we will refer to the disjoint classes as:*

$$\mathcal{S}_{\mathrm{Bnd}} = \{x | x \in \mathcal{D}_{\mathrm{test}}, T_{\mathrm{Bnd}}(x) > \tau_{\mathrm{Bnd}}, T_{\mathrm{IDM}}(x) \leq \tau_{\mathrm{IDM}}\}$$
$$\mathcal{S}_{\mathrm{IDM}} = \{x | x \in \mathcal{D}_{\mathrm{test}}, T_{\mathrm{Bnd}}(x) \leq \tau_{\mathrm{Bnd}}, T_{\mathrm{IDM}}(x) > \tau_{\mathrm{IDM}}\}$$
$$\mathcal{S}_{\mathrm{B\&I}} = \{x | x \in \mathcal{D}_{\mathrm{test}}, T_{\mathrm{Bnd}}(x) > \tau_{\mathrm{Bnd}}, T_{\mathrm{IDM}}(x) > \tau_{\mathrm{IDM}}\}$$

Test examples that are flagged by uncertainty method $u$—yet do not meet any of the thresholds—are marked as *Other*.

**In a nutshell** DAUC uses the OOD, IDM and Bnd classes to categorize model uncertainty—see Table 2 for an overview. Better predictions may be possible for IDM samples, in case a different classifier is used. For samples that are also labelled as Bnd, fine-tuning the existing model—possibly after gathering more data—may be able to separate the different classes better. IDM samples that are not in the Bnd class are harder, and may only be classified correctly if a different latent representation is found, or an different training set is used. We explore the idea of improving the performance on uncertain examples in Appendix B. In Section 4.1 we explore the distinction between classes further.

Table 2: Summary of different uncertainty types

| Type | Definition | Description |
|------|-----------|-------------|
| OOD | $1/p(x\|\mathcal{D}_{\mathrm{train}}) > \tau_{\mathrm{OOD}}$ | Samples that do not resemble the training data. Additional labelled data that covers this part of the input space is required to improve performance on these samples. |
| Bnd | $\sum_{c \neq \hat{c}} P_{c,c}(x\|\mathcal{D}_{\mathrm{val}}) > \tau_{\mathrm{Bnd}}$ | Samples near the boundaries in the latent space. Predictions on these samples are sensitive to small changes in the predictor, and fine-tuning the prediction model may yield better predictions. |
| IDM | $\sum_{c \neq \hat{c}} P_{c,\hat{c}}(x\|\mathcal{D}_{\mathrm{val}}) > \tau_{\mathrm{IDM}}$ | Samples that are likely to be misclassified, since similar examples were misclassified in the validation set. |

## 4 Experiments

In this section, we demonstrate our proposed method with empirical studies. Specifically, we use two experiments as **Proof-of-Concept**, and two experiments as **Use Cases**. Specifically, in Sec. 4.1 we visualize the different classes of flagged examples on a **modified Two-Moons** dataset; in Sec. 4.2 we quantitatively assess DAUC's categorization accuracy on the **Dirty-MNIST dataset** [41]; in Sec. 4.3, we present a use case of DAUC—comparing existing uncertainty estimation benchmarks; in Sec. 4.4, we demonstrate another important use case of DAUC—improving uncertain predictions.

Our selection of the Dirty-MNIST dataset for empirical evaluation was motivated by the pursuit of better reproducibility. As an existing publicly available resource, Dirty-MNIST provides gold labels for boundary classes and OOD examples, making it particularly suitable for benchmarking the performance of DAUC.

Recognizing the importance of demonstrating the broader applicability of DAUC, we have extended our evaluation to include results on the Dirty-CIFAR dataset. Details of this additional evaluation are available in Appendix C.4, where we also describe how we created the dataset. This dataset will also be made publicly available. Additional empirical evidence that justifies DAUC is provided in Appendix C.

### 4.1 Visualizing DAUC with Two-Smiles

#### 4.1.1 Experiment Settings

To highlight the different types of uncertainty, we create a modified version of the Two-Moons dataset, which we will call "Two-Smiles". We use scikit-learn's *datasets* package to generate 6000 two-moons examples with a noise rate of $0.1$. In addition, we generate two Gaussian clusters of 150 examples centered at $(0, 1.5)$ and $(1, -1)$ for training, validation and test, and mark them as the positive and negative class separately. The data is split into a training, validation and test set. We add an additional 1500 OOD examples to the test set, that are generated with a Gaussian distribution centered at $(2, 2)$ and $(-1, -1.5)$. Overall, the test set counts 4500 examples, out of which 1500 are positive examples, 1500 are negative examples and 1500 are OOD—see Figure 2 (a).

#### 4.1.2 Results

We demonstrate the intuition behind the different types of uncertainty, by training a linear model to classify the Two-Smiles data—i.e. using a composition of an identity embedding function and linear classifier, see Assumption 1. Figure 2 (b) shows the test-time misclassified examples. Since the identity function does not linearly separate the two classes and outliers are unobserved at training



| (a) Test Dataset | (b) Test Errors | (c) Flagged OOD | (d) Flagged Bnd | (e) Flagged IDM |

Figure 2: Visualization of our proposed framework on the Two-Smiles dataset with a linear model.

time, the classifier makes mistakes near class boundaries, in predicting the OOD examples, as well as the clusters at $(0, 1.5)$ and $(1, -1)$. In Figure 2 (c)-(e), we see that DAUC correctly flags the OOD examples, boundary examples and the IDM examples. Also note the distinction between $\mathcal{S}_{\text{B&I}}$ and $\mathcal{S}_{\text{IDM}}$. Slightly fine-tuning the prediction model may yield a decision boundary that separates the $\mathcal{S}_{\text{B&I}}$ samples from the incorrect class. On the other hand, this will not suffice for the $\mathcal{S}_{\text{IDM}}$ examples—i.e. the clusters at $(0, 1.5)$ and $(1, -1)$—which require a significantly different representation/prediction model to be classified correctly.

## 4.2 Verifying DAUC with Dirty-MNIST:

### 4.2.1 Experiment Settings

We evaluate the effectiveness of DAUC on the Dirty-MNIST dataset [41]. The training data set is composed of the vanilla MNIST dataset and Ambiguous-MNIST, containing ambiguous artificial examples—e.g., digits 4 that look like 9s. The test set is similar but also contains examples from Fashion-MNIST as OOD examples. The advantage of this dataset is that the different data types roughly correspond to the categories we want to detect, and as such we can verify whether DAUC's classification corresponds to these "ground-truth" labels. Specifically, we deem all examples from MNIST to be non-suspicious, associate Ambiguous-MNIST with class Bnd, and associate Fashion-MNIST with class OOD (There are $1,000$ OOD examples out of the $11,000$ test examples). See [41] for more details on the dataset. We use a 3-layer CNN model with ReLU activation and MaxPooling as the backbone model in this section.

### 4.2.2 Results

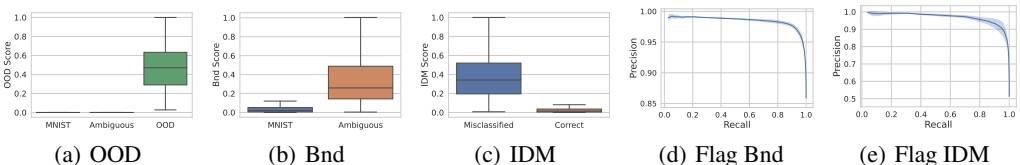

| (a) OOD | (b) Bnd | (c) IDM | (d) Flag Bnd | (e) Flag IDM |

Figure 3: (a-c) Box-plots of different data classes and corresponding scores. All values are scaled by being divided by the maximal value. (d-e) Precision-Recall curves for different choices of thresholds in flagging Bnd examples and IDM examples.

**Flagging Outliers** We assume the ground-truth label of all Fashion-MNIST data is OOD. Comparing these to the classification by DAUC, we are able to quantitatively evaluate the performance of DAUC in identifying outliers. We compute precision, recall and F1-scores for different classes, see Figure 3 (a) and Table 3. All Fashion-MNIST examples are successfully flagged as OOD by DAUC. In Appendix C.3 we include more OOD experiments, including comparison to OOD detector baselines.

Table 3: Quantitative results on the Dirty-MNIST dataset. Our proposed method can flag all three classes of uncertain examples with high F1-Score. The results presented in the table are based on 8 repeated runs.

| Category | Precision | Recall | F1-Score |
|----------|-----------|--------|----------|
| OOD | $1.000 \pm 0.000$ | $1.000 \pm 0.000$ | $1.000 \pm 0.000$ |
| Bnd | $0.959 \pm 0.008$ | $0.963 \pm 0.006$ | $0.961 \pm 0.003$ |
| IDM | $0.918 \pm 0.039$ | $0.932 \pm 0.024$ | $0.924 \pm 0.012$ |

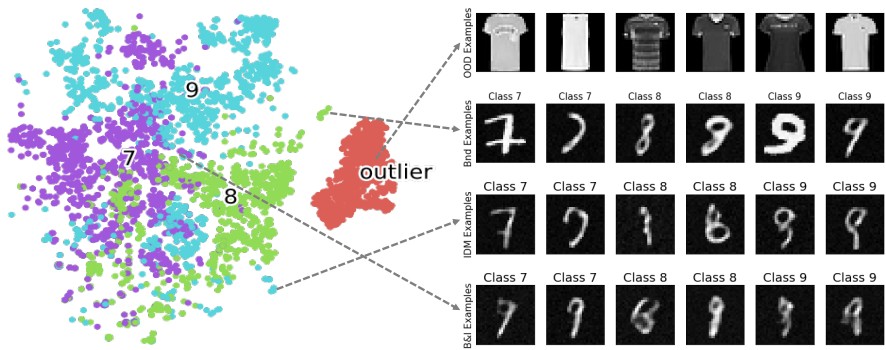

Figure 4: Examples of different uncertainty classes $\mathcal{S}_{\text{OOD}}$, $\mathcal{S}_{\text{IDM}}$, $\mathcal{S}_{\text{Bnd}}$ and $\mathcal{S}_{\text{B\&I}}$. For better visualization, we only plot some classes including the outliers. t-SNE [42] is leveraged in generating low-dim visualizations.

**Flagging Bnd Examples**    We expect most of the boundary examples to belong to the Ambiguous-MNIST class, as these have been synthesised using a linear interpolation of two different digits in latent space [41]. Figure 3 (b) shows that DAUC's boundary scores are indeed significantly higher in Ambiguous-MNIST compared to vanilla MNIST. Figure 3 (d) shows the precision-recall curve of DAUC, created by varying threshold $\tau_{\text{Bnd}}$. Most boundary examples are correctly discovered under a wide range of threshold choices. This stability of uncertainty categorization is desirable, since $\tau_{\text{Bnd}}$ is usually unknown exactly.

**Flagging IDM Examples**    In order to quantitatively evaluate the performance of DAUC on flagging IDM examples, we use a previously unseen hold-out set, which is balanced to consist of $50\%$ misclassified and $50\%$ correctly classified examples. We label the former as test-time IDM examples and the latter as non-IDM examples, and compare this to DAUC's categorization. Figure 3c shows DAUC successfully assigns significantly higher IDM scores to the examples that are to-be misclassified.

Varying $\tau_{\text{IDM}}$ we create the precision-recall curve for the IDM class in Figure 3 (e), which is fairly stable w.r.t. the threshold $\tau_{\text{IDM}}$. In practice, we recommend to use the prediction accuracy on the validation dataset for setting $\tau_{\text{IDM}}$—see Appendix A.2.

**Visualizing Different Classes**    Figure 4 shows examples from the $\mathcal{S}_{\text{OOD}}$, $\mathcal{S}_{\text{Bnd}}$, $\mathcal{S}_{\text{IDM}}$ and $\mathcal{S}_{\text{B\&I}}$ sets. The first row shows OOD examples from the Fashion-MNIST dataset. The second row shows boundary examples, most of which indeed resemble more than one class. The third row shows IDM examples, which DAUC thinks are likely to be misclassified since mistakes were made nearby on the validation set. Indeed, these examples look like they come from the "dirty" part of dirty-MNIST, and most digits are not clearly classifiable. The last row contains B&I examples, which exhibit both traits.

### 4.3    Benchmark Model Uncertainty Categorization

In this section, we demonstrate how DAUC categorizes existing UQ model uncertainty. We compare UQ methods MC-Dropout [10] (**MCD**), Deep-Ensemble [6] (**DE**) and Bayesian Neural Networks [9] (**BNNs**). These methods output predictions and uncertainty scores simultaneously, we follow the traditional approach to mark examples as uncertain or trusted according to their uncertain scores and specified thresholds. To demonstrate how DAUC categorizes all ranges of uncertain examples, we present results from top $5\%$ to the least $5\%$ uncertainty.

Figure 5 compares the proportion of different classes of flagged examples across the three UQ methods. The first and second rows show the *total number* and *proportion* of flagged examples for each class, respectively. There is a significant difference in what the different UQ methods identify. There is a significant difference between types of classes that the different UQ methods identify as discussed in the literature [19–21], which have shown that some UQ methods might not yield high quality uncertainty estimates due to the learning paradigm or sensitivity to parameters. Let us look at each column more carefully:

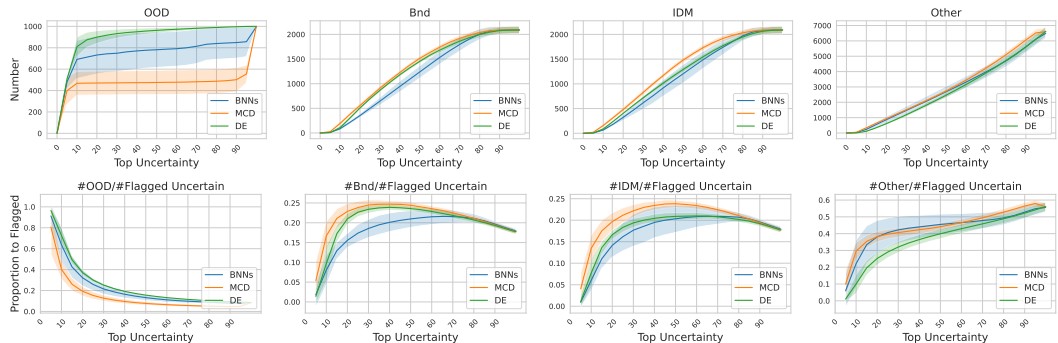

Figure 5: Results of applying our method in categorizing different uncertainty estimation methods. First row: comparisons on the numbers in different classes of examples. Second row: comparisons on the proportion of different classes of flagged examples to the total number of identified uncertain examples. Different methods tend to identify different certain types of uncertain examples. The results presented are based on 8 repeated runs with different random seeds.

1. DE tends to identify the OOD examples as the most uncertain examples. Specifically, looking at the bottom figure we see that the top $5\%$ untrusted examples identified by DE are almost all OOD examples, which is not the case for the other UQ methods. By contrast, MCD is poor at identifying OOD examples; it flags some of the OOD samples as the most certain. This is explained by MCD's mechanism. Uncertainty is based on the difference in predictions across different drop-outs, however this could lead to outliers always having the same prediction— due to correlation between different nodes in the MCD model, extreme values of the OOD examples may always saturate the network and lead to the same prediction, even if some nodes are dropped out. DE is most apt at flagging the OOD class. This confirms the finding by [20] who showed that DE outperforms other methods under dataset shift—which is effectively what OOD represents.

2. After the OOD examples have been flagged, the next most suspicious examples are the Bnd and IDM classes—see columns 2 and 3. The number of these examples increases almost linearly with the number of flagged examples, until at about $88\%$ no more examples are flagged as IDM and Bnd. This behaviour is explained by the Vanilla MNIST examples— which are generally distinguishable and relatively easily classified correctly—accounting for about $15\%$ of the test examples.

3. As expected, the number of examples belonging to the *Other* class increases when more examples are flagged as uncertain. This makes sense, as the *Other* class indicates we cannot flag why the methods flagged these examples as uncertain, i.e. maybe these predictions should in fact be trusted.

## 4.4 Improving Uncertain Predictions

In this section, we explore the inverse direction, i.e. employing DAUC's categorization for creating better models. We elaborate the practical method in Appendix B. We experiment on UCI's **Covtype**, **Digits** and **Spam** dataset [43] with linear models (i.e. $g = Id$) and experiment on **DMNIST** with ResNet-18 learning the latent representation.

Our empirical studies (Figure 8, Appendix C) have shown that the B&I examples are generally hardest to classify, hence we demonstrate the use case of DAUC for improving the predictive performance on this class. We vary the proportion $q$ of training samples that we discard before training new prediction model $f_{\text{B\&I}}$—only saving the training samples that resemble the B&I dataset most—see Figure 6. We find that retraining the linear model with filtered training data according to Eq. 6 significantly improves the performance. We observe that performance increases approximately linearly proportional to $q$, until the amount of training data becomes too low. The latter depends on the dataset and model used.

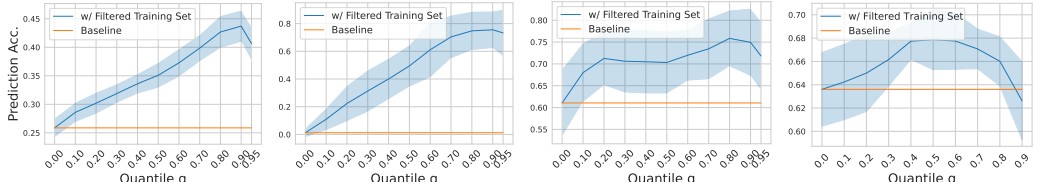

Figure 6: Improved uncertain predictions. Left to right: **Covtype**, **Digits**, **Spam**, **DMNIST**. Experiment are performed with different proportion of training samples discarded: e.g., with $q = 0.0$, all examples in the training set are used; while with $q = 0.9$, only top $10\%$ examples most resembling the test data are used for training. The results presented are based on 10 repeated runs with different random seeds.

## 5    Conclusion and Future Work

We have proposed DAUC, a framework for model uncertainty categorization. DAUC categorizes uncertain examples identified by UQ benchmarks into three classes—OOD, Bnd and IDM. These classes correspond to different causes for the uncertainty and require different strategies for possibly better predictions. We have demonstrated the power of DAUC by inspecting three different UQ methods—highlighting that each one identifies different examples. We believe DAUC can aid the development and benchmarking of UQ methods, paving the way for more trustworthy ML models.

In future work, DAUC has great potential to be extended to more general tasks, such as the regression setting, and reinforcement learning setting, where uncertainty quantification is essential. The idea of separating the source of uncertainty improves not only exploration [44] but also exploitation in the offline settings [30–35].

In the era of Large Language Models (LLMs) [45, 46], uncertainty quantification is essential in evaluating the task performance of LLMs [47], and holds great potential for AI alignment [48, 49] — as understanding the ability boundary of LLMs is essential, and identifying the suspicious outputs of LLMs can be potentially addressed by extending the framework of DAUC to the LLMs' setting.

## Acknoledgement

HS acknowledges and thanks the funding from the Office of Naval Research (ONR). We thank the van der Schaar lab members for reviewing the paper and sharpening the idea. We thank all anonymous reviewers, ACs, SACs, and PCs for their efforts and time in the reviewing process and in improving our paper.

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

# A Missing Details

## A.1 Motivations for working with model latent space

In Section 3, we introduced the confusion density matrix that allows us to categorize suspicious examples at testing time. Crucially, this density matrix relies on kernel density estimations in the latent space $\mathcal{H}$ associated with the model $f$ through Assumption 1. Why are we performing a kernel density estimation in latent space rather than in input space $\mathcal{X}$? The answer is fairly straightforward: we want our density estimation to be coupled to the model and its predictions.

Let us now make this point more rigorous. Consider two input examples $x_1, x_2 \in \mathcal{X}$. The model assigns a representations $g(x_1), g(x_2) \in \mathcal{H}$ and class probabilities $f(x_1), f(x_2) \in \mathcal{Y}$. If we define our kernel $\kappa$ in latent space $\mathcal{H}$, this often means[2] that $\kappa[g(x_1), g(x_2)]$ grows as $\|g(x_1) - g(x_2)\|_{\mathcal{H}}$ decreases. Hence, examples that are assigned a similar latent representation by the model $f$ are related by the kernel. Since our whole discussion revolves around model *predictions*, we would like to guarantee that two examples related by the kernel are given similar predictions by the model $f$. In this way, we would be able to interpret a large kernel density $\kappa[g(x_1), g(x_2)]$ as a hint that the predictions $f(x_1)$ and $f(x_2)$ are similar. We will now show that, under Assumption 1, such a guarantee exists. Similar to [50], we start by noting that

$$\|(l \circ g)(x_1) - (l \circ g)(x_2)\|_{\mathbb{R}^C} = \|l\,[g(x_1) - g(x_2)]\|_{\mathbb{R}^C}$$
$$\leq \|l\|_{\mathrm{op}} \|g(x_1) - g(x_2)\|_{\mathcal{H}},$$

where $\|\cdot\|_{\mathbb{R}^C}$ is a norm on $\mathbb{R}^C$ and $\|l\|_{\mathrm{op}}$ is the operator norm of the linear map $l$. In order to extend this inequality to black-box predictions, we note that the normalizing map in Assumption 1 is often a Lipschitz function with Lipschitz constant $\lambda \in \mathbb{R}$. For instance, a Softmax function with inverse temperature constant $\lambda^{-1}$ is $\lambda$-Lipschitz [51]. We use this fact to extend our inequality to predicted class probabilities:

$$\|f(x_1) - f(x_2)\|_{\mathcal{Y}} = \|(\varphi \circ l \circ g)(x_1) - (\varphi \circ l \circ g)(x_2)\|_{\mathcal{Y}}$$
$$\leq \lambda \|(l \circ g)(x_1) - (l \circ g)(x_2)\|_{\mathbb{R}^C}$$
$$\leq \lambda \|l\|_{\mathrm{op}} \|g(x_1) - g(x_2)\|_{\mathcal{H}}.$$

This crucial inequality guarantees that examples $x_1, x_2 \in \mathcal{X}$ that are given a similar latent representation $g(x_1) \approx g(x_2)$ will also be given a similar prediction $f(x_1) \approx f(x_2)$. In short: two examples that are related according to a kernel density defined in the model latent space $\mathcal{H}$ are guaranteed to have similar predictions. This is the motivation we wanted to support the definition of the kernel $\kappa$ in latent space.

An interesting question remains: is it possible to have similar guarantees if we define the kernel in input space? When we deal with deep models, the existence of adversarial examples indicates the opposite [52]. Indeed, if $x_2$ is an adversarial example with respect to $x_1$, we have $x_1 \approx x_2$ (and hence $\|x_1 - x_2\|_{\mathcal{X}}$ small) with two predictions $f(x_1)$ and $f(x_2)$ that are significantly different. Therefore, defining the kernel $\kappa$ in input space might result in relating examples that are given a significantly different prediction by the model. For this reason, we believe that the latent space is more appropriate in our setting.

## A.2 Details: Flagging IDM and Bnd Examples with Thresholds

In order to understand uncertainty, it will be clearer to map those scores into binary classes with thresholds. In our experiments, we use empirical quantiles as thresholds. e.g., to label an example as IDM, we specify an empirical quantile number $q$, and calculate the corresponding threshold based on the order statistics of IDM Scores for test examples: $S_{\mathrm{IDM}}^{(1)}, ..., S_{\mathrm{IDM}}^{(|\mathcal{D}_{\mathrm{test}}|)}$, where $S_{\mathrm{IDM}}^{(n)}$ denotes the $n$-th smallest IDM score out of $|\mathcal{D}_{\mathrm{test}}|$ testing-time examples. Then, the threshold given quantile number $q$ is

$$\tau_{\mathrm{IDM}}(q) \equiv S_{\mathrm{IDM}}^{(\lfloor |\mathcal{D}_{\mathrm{test}}| \cdot q \rfloor)}.$$

---

[2]This is the case for all the kernels that rely on a distance (e.g. the Radial Basis Function Kernel, the Matern kernel or even Polynomial kernels [39]).

Similarly, we can define quantile-based threshold in flagging Bnd examples based on the order statistics of Bnd Scores for test examples, such that for given quantile $q$,

$$\tau_{\text{Bnd}}(q) \equiv S_{\text{Bnd}}^{(\lfloor |\mathcal{D}_{\text{test}}| \cdot q \rfloor)}.$$

Practically, a natural choice of $q$ is to use the validation accuracy: when there are $1 - q$ examples misclassified in the validation set, we also expect the testing-time in distribution examples with the highest $1 - q$ to be marked as Bnd or IDM examples.

# B    Improving Predicting Performance of Uncertain Examples

Knowing the category that a suspicious example belongs to, can we improve its prediction? For ease of exposition, we focus on improving predictions for $\mathcal{S}_{\text{B\&I}}$.

Let $p\left(x \mid \mathcal{S}_{\text{B\&I}}\right)$ be the latent density be defined as in Definition 1. We can improve the prediction performance of the model on $\mathcal{S}_{\text{B\&I}}$ examples by focusing on the part of examples in the training set that are closely related to those suspicious examples. We propose to refine the training dataset $\mathcal{D}_{\text{train}}$ by only keeping the examples that resembles the latent representations for the specific type of test-time suspicious examples, and train another model on this subset of the training data:

$$\tilde{\mathcal{D}}_{\text{train}} \equiv \{x \in \mathcal{D}_{\text{train}} | p\left(x \mid \mathcal{S}_{\text{B\&I}}\right) \geq \tau_{\text{test}}\}, \tag{6}$$

where $\tau_{\text{test}}$ is a threshold that can be adjusted to keep a prespecified proportion $q$ of the related training data. Subsequently, new prediction model $f_{\text{B\&I}}$ is trained on $\tilde{\mathcal{D}}_{\text{train}}$.

Orthogonal to ensemble methods that require multiple models trained independently, and improve *overall* prediction accuracy by bagging or boosting, our method is targeted at improving the model's performance on a *specified* subclass of test examples by finding the most relevant training examples. Our method is therefore more transparent and can be used in parallel with ensemble methods if needed.

**Threshold** $\tau_{\text{test}}(q)$    For every training example $x \in \mathcal{D}_{\text{train}}$, we have the latent density $p(x | \mathcal{D}_{\text{B\&I}})$ over the B&I class of the test set. With their order statistics $p_{(1)}(x | \mathcal{D}_{\text{B\&I}}), ..., p_{(|\mathcal{D}_{\text{train}}|)}(x | \mathcal{D}_{\text{B\&I}})$. Given quantile number $q$, our empirical quantile based threshold $\tau_{\text{test}}$ is chosen as

$$\tau_{\text{test}}(q) \equiv p_{(\lfloor q \cdot |\mathcal{D}_{\text{train}}| \rfloor)}(x | \mathcal{D}_{\text{B\&I}}).$$

During the inverse training time, we train our model to predict those B&I class of test examples only with the training data with higher density than $\tau_{\text{test}}(q)$. We experiment with different choices of $q$ in the experiment (Figure 6 in Sec. 4.4).

# C    Additional Experiments

## C.1    Categorization of Uncertainty under Different Thresholds

In the main text, we provide results with $\tau_{\text{Bnd}} = \tau_{\text{IDM}} = 0.8$, which approximates the accuracy on validation set—as a natural choice. In this section, we vary these thresholds and show in Figure 7 that changing those thresholds does not significantly alter the conclusions drawn above.

Figure 8 looks more closely into the top $25\%$ uncertain examples for each method, and the accuracy on each of the uncertainty classes. As expected, the accuracy of the B&I examples is always lower than that of the trusted class, meaning that those examples are most challenging for the classifier. And the accuracy of flagged classes are always lower than the *other* class, verifying the proposed categorization of different classes.

## C.2    Inverse Direction: More Results

In the main text, we show the results on improving prediction performance on the B&I class with training example filtering (On the Covtype, Digits dataset). More results on other classes of examples are provided in this section.

We experiment on three UCI datasets: **Covtype**, **Digits**, and **Spam**. And experiment with three classes we defined in this work:

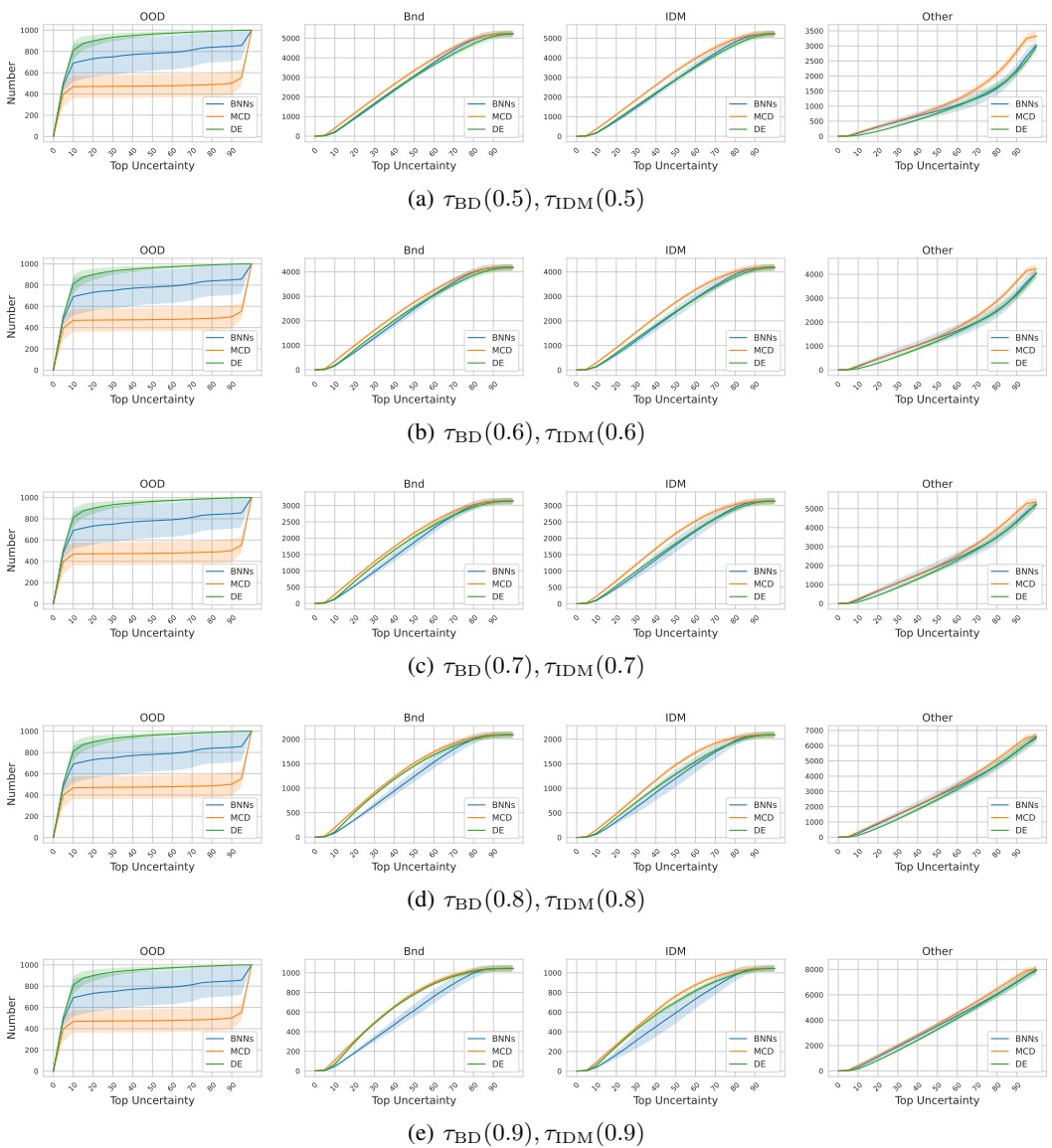

Figure 7: Experiments on different choices of thresholds. The results presented are based on 8 repeated runs with different random seeds.

1. **B&I** class (Figure 9). As we have discussed in our main text, the prediction accuracy on the B&I class are always the lowest among all classes. By training with filtered examples in $\mathcal{D}_{\text{train}}$ rather than the entire training set, the B&I class of examples can be classified with a remarkably improved accuracy.

2. **Bnd** class (Figure 10). This class of examples are located at boundaries in the latent space of validation set, but not necessarily have been misclassified. Therefore, their performance baseline (training with the entire $\mathcal{D}_{\text{train}}$) is relatively high. The improvement is clear but not as much as on the other two classes.

3. **IDM** class (Figure 11). For this class of examples, similar mistakes have been make in the validation set, yet those examples are not necessarily located in the boundaries—the misclassification may be caused by ambiguity in decision boundary, imperfectness of either the model or the dataset. The primal prediction accuracy on this class of examples is lower than the Bnd class but higher than the B&I class, training with filtered $\mathcal{D}_{\text{train}}$ also clearly improve the performance on this class of examples.

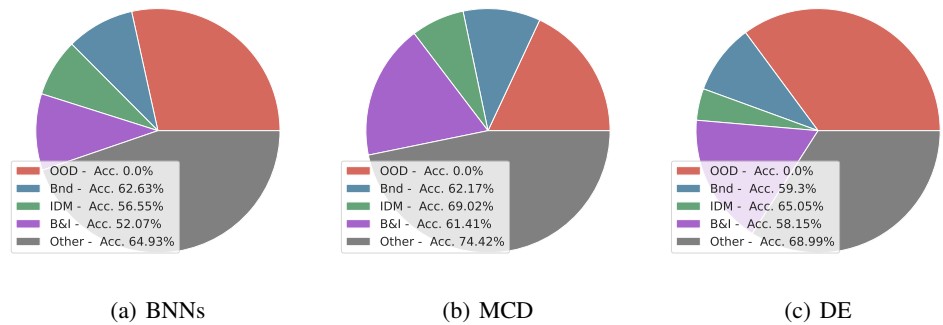

(a) BNNs         (b) MCD         (c) DE

Figure 8: The top 25% uncertain examples identified by different methods. Legend of each figure provide the accuracy and proportion of each class. As the classifier can not make correct predictions on the OOD examples, it's always better for uncertainty estimators to flag more OOD examples.

Table 4: DAUC is not the only choice in identifying OOD examples. On the Dirty-MNIST dataset, DAUC, Outlier-AE and the IForest can identify most outliers in the test dataset. (Given threshold = 1.0 for those two benchmark methods).

| Method | Precision | Recall | F1-Score |
|---|---|---|---|
| DAUC | $1.0000 \pm 0.0000$ | $1.0000 \pm 0.0000$ | $1.0000 \pm 0.0000$ |
| Outlier-AE | $1.0000 \pm 0.0000$ | $1.0000 \pm 0.0000$ | $1.0000 \pm 0.0000$ |
| IForest [54] | $0.9998 \pm 0.0004$ | $1.0000 \pm 0.0000$ | $0.9999 \pm 0.0002$ |

## C.3 Alternative Approach in Flagging OOD

As we have mentioned in the main text, although DAUC has a unified framework in understanding all three types of uncertainty the uncertainty caused by OOD examples can also be identified by off-the-shelf algorithms. We compare DAUC to two existing outlier detection methods in Table 4, where all methods achieve good performance on the Dirty-MNIST dataset. Our implementation is based on Alibi Detect [53].

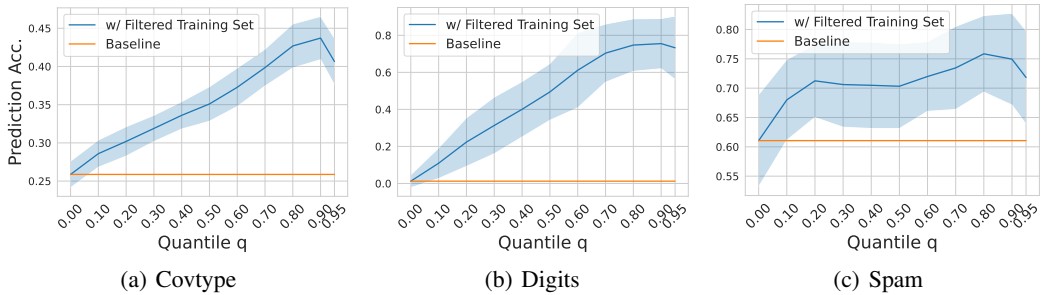

(a) Covtype         (b) Digits         (c) Spam

Figure 9: Experiments on the B&I class (reported in the main text). The results presented are based on 10 repeated runs with different random seeds.

## C.4 Experiments on Dirty-CIFAR-10

**Dataset Discription** In this experiment, we introduce a revised version of the CIFAR-10 dataset to test DAUC's scalability. Similar to the Dirty-MNIST datset [41], we use linear combinations of the latent representation to construct the "boundary" class. In the original CIFAR-10 Dataset, each of the 10 classes of objects has 6000 training examples. We split the training set into training set (40%), validation set (40%) and test set (20%). To verify the performance of DAUC in detecting OOD examples, we randomly remove one of those 10 classes (denoted with class-$i$) during training

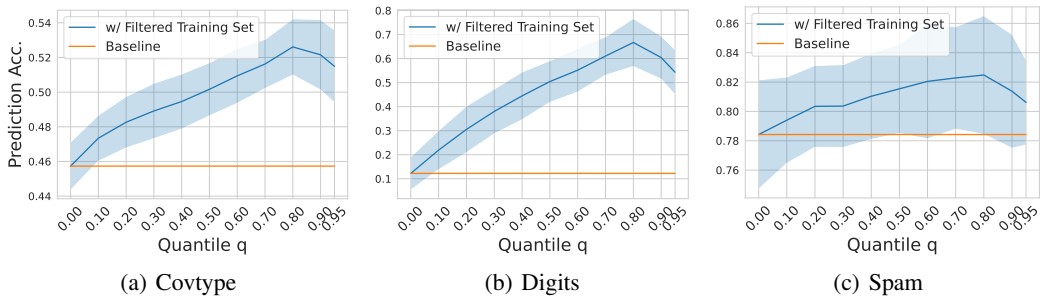

(a) Covtype       (b) Digits       (c) Spam

Figure 10: Experiments on the Bnd class. The results presented are based on 10 repeated runs with different random seeds.

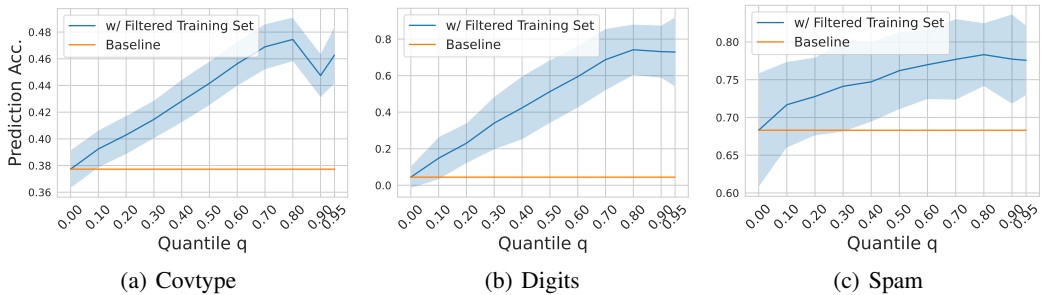

(a) Covtype       (b) Digits       (c) Spam

Figure 11: Experiments on the IDM class. The results presented are based on 10 repeated runs with different random seeds.

and manually concatenate OOD examples with the test dataset, with label $i$. In our experiment, we use 1000 MNIST digits as the OOD examples, with zero-padding to make those digits share the same input shape as the CIFAR-10 images. Combining those boundary examples, OOD examples and the vanilla CIFAR-10 examples, we get a new benchmark, dubbed as Dirty-CIFAR-10, for quantitative evaluation of DAUC.

**Quantify the performance of DAUC on Dirty-CIFAR-10**     Quantitatively, we evaluate the performance of DAUC in categorizing all three classes of uncertain examples. Results of averaged performance and standard deviations based on $8$ repeated runs are provided in Table 5.

Table 5: Quantitative results on the Dirty-CIFAR-10 dataset. DAUC scales well and is able to categorize all three classes of uncertain examples. Results presented in the table are based on 8 repeated runs with different random seeds.

| Category | Precision | Recall | F1-Score |
|----------|-----------|--------|----------|
| OOD | $0.986 \pm 0.003$ | $0.959 \pm 0.052$ | $0.972 \pm 0.027$ |
| Bnd | $0.813 \pm 0.002$ | $0.975 \pm 0.000$ | $0.887 \pm 0.001$ |
| IDM | $0.688 \pm 0.041$ | $0.724 \pm 0.017$ | $0.705 \pm 0.027$ |

**Categorize Uncertain Predictions on Dirty-CIFAR-10**     Similar to Sec. 4.3 and Figure 5, we can categorize uncertain examples flagged by BNNs, MCD, and DE using DAUC—see Figure 12. We find that in the experiment with CIFAR-10, DE tends to discover more OOD examples as top uncertain examples. Differently, although BNNs flag fewer OOD examples as top-uncertain, they continuously discover those OOD examples and are able to find most of them for the top $50\%$ uncertainty. On the contrary, MCD performs the worst among all three methods, similar to the result is drawn from the DMNIST experiment. On the other hand, while BNN is good at identifying OOD examples, it flags less uncertain examples in the Bnd and IDM classes. DE is the most apt at flagging both Bnd and

IDM examples and categorizes far fewer examples into the *Other* class. **These observations are well aligned with the experiment results we had with DMNIST in Sec. 4.3, showing the scalability of DAUC to large-scale image datasets.**

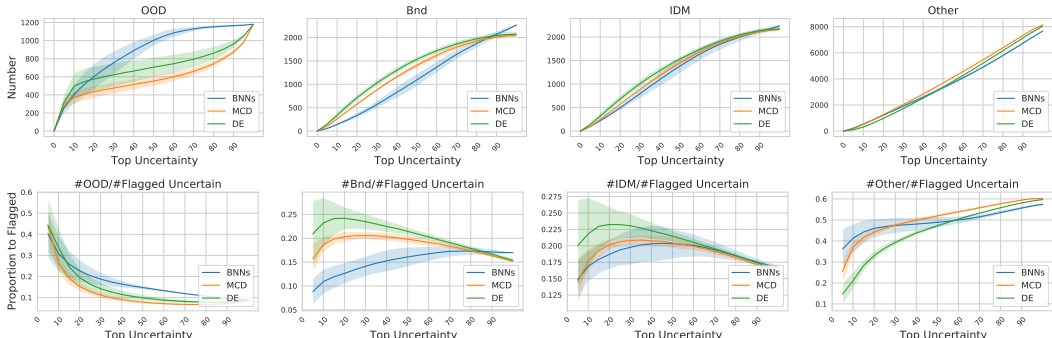

Figure 12: Experiments on the CIFAR-10 dataset. Results of applying DAUC in categorizing different uncertainty estimation methods. First row: comparisons of the numbers in different classes of examples. Second row: comparisons on the proportion of different classes of flagged examples to the total number of identified uncertain examples. Different methods tend to identify different certain types of uncertain examples. The results presented are based on 8 repeated runs with different random seeds.

# D    Implementation Details

## D.1    Code

Our code is available at `https://github.com/vanderschaarlab/DAUC`.

## D.2    Hyperparameters

### D.2.1    Bandwidth

In our experiments, we use (z-score) normalized latent representations and bandwidth $1.0$. In the inverse direction, as the sample sizes are much smaller, a bandwidth of $0.01$ is used as the recommended setting. There is a vast body of research on selecting a good bandwidth for Kernel Density Estimation models [55–57] and using these to adjust DAUC's bandwidth to a more informed choice may further improve performance.

## D.3    Inverse Direction: Quantile Threshold q

As depicted in Appendix A.2, a natural choice of $q$ is to use the validation accuracy. We use this heuristic approach in our experiments for the inverse direction.

## D.4    Model Structure

In our experiments, we implement **MCD** and **DE** with 3-layer-CNNs with ReLU activation. Our experiments on **BNNs** are based on the IBM UQ360 software [58]. More details of the convolutional network structure are provided in Table 6.

## D.5    Implementation of Kernel Density Estimation and Repeat Runs

Our implementation of KDE models is based on sklearn's KDE package [59]. Gaussian kernels are used as default settings. We experiment with 8-10 random seeds and report the averaged results and standard deviations. In our experiments, we find using different kernels in density estimation provides highly correlated scores. We calculate the Spearman's $\rho$ correlation between scores DAUC gets over 5 runs with Gaussian, Tophat, and Exponential kernels under the same bandwidth. Changing the kernel brings highly correlated scores (all above $\mathbf{0.86}$) for DAUC and, hence, has a minor impact on

Table 6: Network Structure

| Layer | Unit | Activation | Pooling |
|-------|------|------------|---------|
| Conv 1 | $(1, 32, 3, 1, 1)$ | ReLU() | MaxPool2d(2) |
| Conv 2 | $(32, 64, 3, 1, 1)$ | ReLU() | MaxPool2d(2) |
| Conv 3 | $(64, 64, 3, 1, 1)$ | ReLU() | MaxPool2d(2) |
| FC | $(64 \times 3 \times 3, 40)$ | ReLU() | - |
| Out | $(40, N_{\text{Class}})$ | SoftMax() | - |

DAUC's performance. We preferred KDE since the latent representation is relatively low-dimensional. We found that a low-dim latent space (e.g., 10) works well for all experiments (including CIFAR-10).

### D.6 Hardware

All results reported in our paper are conducted with a machine with 8 Tesla K80 GPUs and 32 Intel(R) E5-2640 CPUs. The computational cost is mainly in density estimation, and for low-dim representation space, such an estimation can be efficient: running time for DAUC on the Dirty-MNIST dataset with KDE is approximately 2 hours.

## Assumptions and Limitations

In this work, we introduced the confusion density matrix that allows us to categorize suspicious examples at testing time. Crucially, this density matrix relies on kernel density estimations in the latent space $\mathcal{H}$ associated with the model $f$ through Assumption 1. We note this assumption generally holds for most modern uncertainty estimation methods.

While the core contribution of this work is to introduce the concept of confusion density matrix for uncertainty categorization, the density estimators leveraged in the latent space can be further improved. We leave this to future work.

## Broader Impact

While previous works on uncertainty quantification (UQ) focused on the discovery of uncertain examples, in this work, we propose a practical framework for categorizing uncertain examples that are flagged by UQ methods. We demonstrated that such a categorization can be used for UQ method selection — different UQ methods are good at figuring out different uncertainty sources. Moreover, we show that for the inverse direction, uncertainty categorization can improve model performance.

With our proposed framework, many real-world application scenarios can be potentially benefit. e.g., in Healthcare, a patient marked as uncertain that categorized as OOD — preferably identified by Deep Ensemble, as we have shown — should be treated carefully when applying regular medical experience; and an uncertain case marked as IDM — preferably identified by MCD — can be carefully compared with previous failure cases for a tailored and individualized medication.

