# OpenReview forum: "What is Flagged in Uncertainty Quantification?  Latent Density Models for Uncertainty Categorization"
_NeurIPS.cc/2023/Conference — NeurIPS 2023 poster_

### Official Review · Reviewer_AvJq · 2023-07-03

**Soundness:** 3 good
**Presentation:** 3 good
**Contribution:** 3 good
**Rating:** 6
**Confidence:** 3

**Summary:**

This paper proposes a model-agnostic framework named DAUC to categorize uncertain examples flagged by UQ methods, which introduces the confusion density matrix and provides post-hoc categorization for model uncertainty. Besides, this paper categorizes suspicious examples identified by a given uncertainty method into OOD, Bnd and IDM three classes.

**Strengths:**

1. This paper focuses on samples that remain suspect in uncertain methods, and proposes a model-agnostic density-based approach to mitigate this problem.
2. The overall writing structure of this paper is clear, and the core ideas expressed are clear too.


**Weaknesses:**

1. Why are the uncertain samples of the model divided into three categories? It is only mentioned in the abstract (i.e., a kernel-based approximation of the misclassification density), but there is no relevant description in the paper.
2. The four classes in Figure 1 should be changed to three classes, corresponding to the paper.
3. In the Categorizing Uncertainty part of the Section introduction, it is mentioned that DAUC is a model-agnostic method, but in the experimental part, we do not see the model-agnostic of DAUC.


**Questions:**

1. Checking some details in the paper, such as illustration, etc, corresponding to the paper.
2. Adding the basis of classifying uncertain samples into three categories, or relevant explanations.
3. Adding the explanation of DAUC approach model-agnostic.


**Limitations:**

Is DAUC a model-agnostic approach that can be extended to other tasks or domains?

---

> ### Author Rebuttal · Authors · 2023-08-07
>
> We sincerely appreciate the reviewer's efforts and constructive comments in improving our paper. We will respond to each of the questions in turn:
>
> ---
> ### Q1: The categorization of uncertain examples.
> A1: The categorization of uncertainty into three non-exclusive classes (OOD, Bnd, and IDM) is a fundamental aspect of our approach. These categories enable us to address various types of uncertainty, **each with unique characteristics that can be discerned through the confusion density matrix**.
> - OOD: This class represents examples unlike any seen during training. The kernel density of inference examples with respect to the training dataset identifies this uncertainty. Additional labeled data encompassing this region of the input space would be necessary for performance enhancement on such samples.
> - Bnd: Arising from confusion near decision boundaries, this class is recognized by the diagonal kernel density of inference examples relative to the validation dataset. This form of uncertainty indicates misclassifications stemming from inherent ambiguities in the data. Fine-tuning the prediction model might enhance predictions on these sensitive samples.
> - IDM: This category, identified by the off-diagonal kernel density of inference examples compared to the validation dataset, pertains to the model's intrinsic errors leading to misclassification. Recognizing IDM uncertainty is crucial for understanding the model's inherent biases or limitations.
>
> In a nutshell, these classes correspond to three properties that can be **manifested by the confusion density matrix**: OOD is represented by the **overall density**, Bnd by the **diagonal elements**, and IDM by the **off-diagonal elements**.
>
> To address your concern in Figure 1, we have made it clearer by noting explicitly in our updated manuscript that an example can simultaneously be categorized to the Bnd class and the IDM class, e.g., an example located in the decision boundary can have similar misclassified neighbors. This does not contradict our 3-type categorization according to the confusion density matrix.
>
> We agree that a more extensive explanation would facilitate understanding, and we have therefore **refined and highlighted our introduction to bridge any gap**.
>
> ### Q2: Assumption on model: Model-Agnosticity
> A2: We would start by clarifying the definition of **Model-Agnostic** does not indicate the neural network model used in DAUC and the confusion density matrix. Instead, we would say DAUC is a model-agnostic uncertainty categorization method to highlight its ability to categorize suspicious examples identified by **different UQ models**. The model-agnostic refers to the general applicability of DAUC w.r.t. UQ models, in contrast to neural network models. We illustrate such property by benchmarking different UQ models: MCD, DE, and BNNs.
>
> We have updated our manuscript accordingly to make the definition of Model-Agnosticity in our context clearer.
>
> On the other hand, for the requirement on the neural network model used in DAUC, we would like to emphasize, as highlighted in Table 1, that **DAUC is designed to be flexible in terms of model structure**. It functions effectively under conditions where Assumption 1 can be met, a scenario commonly satisfied when using general neural network architectures.
>
> ### Q3: Other tasks or domains.
> A3: We wish to demonstrate the general applicability of DAUC through the following empirical observations, in addition to the main results presented in the paper.
>
> To demonstrate the broader applicability of DAUC, we **extended our evaluation with the Dirty-CIFAR dataset**. Details of this additional evaluation are available in Appendix C.4, We also described how the dataset is created, and **this dataset will also be made publicly available for future research.**
>
> To stress-test and show the scalability of DAUC, we **experiment with the CIFAR-100 dataset as another additional domain**. Given the limited time and the non-trivial efforts in creating boundary examples and corresponding gold labels, we focus on the inverse direction, as a downstream application, in demonstrating DAUC’s scalability.
>
> The results are shown in **Figure 1 in the attached 1-page PDF file**. For those experiments, we use ResNet-18 to learn representation and identify the IDM and Bnd examples marked as suspicious, and then improve the prediction on those subsets. DAUC scales well on the CIFAR-100 dataset — prediction performance of the classifiers can be improved through filtering similar examples in the training dataset.
>
> To sum up, if we could add up experiments in our main text, appendix, and the updated supplementary response, we used **1 synthetic dataset, 3 tabular datasets, and 3 image datasets in verifying the effectiveness of DAUC**. All of those results support that DAUC provides a new and distinct perspective for identifying uncertainty sources and enables downstream prediction improvement based on the categorization.
>
> ---
> If there are leftover concerns, please let us know and we will do our utmost to address them. Thank you once again for your insightful comments, time, and efforts in reviewing our paper.

---

> > ### Author Response · Authors · 2023-08-19
> > **Dear Reviewer AvJq**
> >
> > Once again, we truly appreciate the time and effort you have devoted to reviewing our paper.
> >
> > As we are approaching the close of the discussion phase, could you please let us know if there are any remaining questions or concerns? We would do our utmost to address them should there be a need!

---

> > ### Comment · Reviewer_AvJq · 2023-08-20
> >
> > Thanks for your detailed response to our concerns. The authors' response addresses most of my concerns, but questions remain about the model-agnostic explanation. The model-agnostic question we raised in the review phase is to ask the authors to reply whether the method proposed in this paper can be inserted into the model related to the task at will. The authors' response seems to misunderstand our meaning. Overall, we still hold our original opinion.

---

> > > ### Author Response · Authors · 2023-08-20
> > > **Thank You for Clarification**
> > >
> > > We deeply appreciate the reviewer for providing further clarification. We would like to take this opportunity and offer a more detailed explanation.
> > >
> > > In our previous response, we focused on explaining DAUC can be used to categorize examples flagged as suspicious by any uncertainty quantification methods. And this is what we mean by "model-agnostic" in our work (Line 26), the "model" in our context refers to the model used in uncertainty quantification.
> > >
> > > > Whether the method proposed in this paper can be inserted into the model related to the task at will?
> > >
> > > For this question, the answer remains affirmative. As a method performing post-hoc uncertainty categorization, DAUC is suitable for any classification model that meets Assumption 1, such as neural networks with a linear output layer. To showcase DAUC's versatility w.r.t. classification models in our experiments, we have used 3-layer CNNs for Dirty-MNIST, MLPs for tabular datasets, and Resnet-18 for the larger-scale CIFAR experiments. DAUC performs well in all those settings.
> > >
> > >
> > > We realize the clarity can be further improved to avoid potential ambiguity. We would like to use **UQ-method-agnostic** in Line 26 in our revision.
> > >
> > > ---
> > > We would appreciate it if the reviewer kindly let us know if our follow-up response could be helpful. In the limited time remaining, we are still eager to do our utmost to address any further questions!
> > >
> > > Many thanks!
> > >
> > > Regards,
> > >
> > > Authors

---

### Official Review · Reviewer_7E4k · 2023-07-04

**Soundness:** 3 good
**Presentation:** 3 good
**Contribution:** 4 excellent
**Rating:** 6
**Confidence:** 4

**Summary:**

In this paper, the authors present a unique approach for classifying uncertainty into distinct categories, providing insights into the reasons behind labeling a particular sample as suspicious or highly uncertain. They develop a kernel density-based confusion density matrix for any neural network which separates suspicious data samples from the uncertainty estimator into three categories: out-of-distribution (OOD), samples near the decision boundary, and misclassified samples. The paper outlines the methodology for constructing this density-based matrix and compares the performance of existing uncertainty estimators using this proposed categorization. Empirical studies and results are provided, demonstrating how this classification can in fact improve the training of predictive models and highlight the reasons for model failure.

**Strengths:**

In my opinion, this categorization process of determining what uncertainties actually flag (OOD, bnd etc.) is the main contribution. Importantly, the paper shows that this is a general framework applicable to any classification model (having a reasonably sized no. of classes). The paper demonstrates experiments on a variety of benchmarks contributing to its efficacy. Also, the paper is well written and simple to follow.

**Weaknesses:**

See questions.

**Questions:**

* In my opinion, at a high level, this method basically computes the kernel similarity scores between the test data and the samples in the train/val distribution in the latent space to find out whether the test sample is OOD, bnd etc. How does the choice of the layer chosen impact the categorization ? Does this change ? This is because, the earlier layers can capture only the basic semantics and the discrepancies between inliers and the suspicious samples may not be captured.

* How exactly is uncertainty from the model computed ? Is epistemic uncertainty alone computed for first filtering the suspicious vs safe samples, or the total uncertainty ? Writing a brief section about the same either in the main/supplement will be beneficial for the reader.

* Can the authors comment on how effective the proposed approach will be to handle open-set data or OOD/anamolous data samples for e.g,  classes unseen during training ? Consider that there is an unc. estimator that is sensitive to open-set data and flags the sample as suspicious. Will the kernel density estimate score that categorizes OOD data in your formulation be able to capture such subtle shifts ? These open-set data need not share the same labels as that of your training data, but share a large amount of semantic information ?

* For computing the confusion density matrix elements for the bnd and IDM samples, how does the properties of the validation data impact the estimation ? For e.g., A certain class c1 can offer no misclassification while another class c2 offers a greater amount of misclassification.

* Can the authors comment on the scalability of the algorithm ? For instance, if the number of classes exceeds K (K>=100),  it is required to obtain a K x K matrix.


**Limitations:**

Yes, the limitations have been listed by the authors in the supplementary.  However, in my opinion,  I am skeptical about the scalability of the approach as well as its ability to handle data from open-set recognition conditions.

---

> ### Author Rebuttal · Authors · 2023-08-07
>
> We deeply appreciate the reviewer's time and effort in evaluating our paper. We will now answer each of the questions:
>
> ---
> ### Q1: Choice of the latent space
> A1: We compute kernel similarity scores within the latent space, as opposed to the input space or basic semantics space, to ensure that the density estimation is directly tied to the model's predictions.
>
> Since our whole discussion revolves around model predictions, we would like to guarantee that two examples related to the kernel are given similar predictions by the model $f$. In this way, we would be able to interpret a large kernel density $\kappa[g(x_1), g(x_2)]$ as a hint that the predictions $f(x_1)$ and $f(x_2)$ are similar. We will now show that, under Assumption 1, such a guarantee exists. First, we note that
>
> $||(l \circ g)(x_1)-(l \circ g)(x_2)||_{\mathbb{R}^C}$
>
> $=||l[g(x_1)-g(x_2)]||_{\mathbb{R}^C}$
>
> $\le||l|| \cdot ||g(x_1)-g(x_2)||_\mathcal{H}$
>
> where $||\cdot||_{\mathbb{R}^C}$ is a norm on $\mathbb{R}^C$ and $||l||$ is the operator norm of the linear map $l$. In order to extend this inequality to black-box predictions, we note that the normalizing map in Assumption 1 is often a Lipschitz function with Lipschitz constant $\lambda \in \mathbb{R}$. For instance, a Softmax function with inverse temperature constant $\lambda^{-1}$ is $\lambda$-Lipschitz. We use this fact to extend our inequality to predicted class probabilities:
>
> $||f(x_1)-f(x_2)||_\mathcal{Y}$
>
> $||\phi\circ l \circ g(x_1)- \phi\circ l \circ g(x_2)||_\mathcal{Y}$
>
> $\le\lambda||l\circ g(x_1)-l\circ g(x_2)||_{\mathbb{R}^C}$
>
> $\le\lambda||l||\cdot||g(x_1)-g(x_2)||_\mathcal{H}$
>
> This crucial inequality guarantees that examples $x_1,x_2\in\mathcal{X}$ that are given a similar latent representation $g(x_1)\approx g(x_2)$ will also be given a similar prediction $f(x_1)\approx f(x_2)$.
>
> In short: **two examples that are related according to a kernel density defined in the model latent space $\mathcal{H}$ are guaranteed to have similar predictions.** This is the motivation we wanted to support the definition of the kernel $\kappa$ in latent space.
> ### Q2: Computation of uncertainty
> A2: DAUC **categorizes the uncertainty** through the confusion density matrix we introduced in Section 3. The categorization does not require any specific uncertainty computation.
>
> In general, DAUC is able to categorize uncertain examples **flagged by any UQ method**. In our paper, we benchmark with MCD, DE and BNNs as 3 typical UQ methods. Given the uncertain scores predicted by those models and thresholds, we are able to mark different proportions of examples as uncertain. To demonstrate how DAUC categorizes all ranges of uncertain examples, we present results on the top $5\%$ to the least $5\%$ uncertainty.
>
> We added an additional section in our appendix to explicitly explain the workflow to improve clarity.
> ### Q3: OpenSet and anomalous detection
> A3: To inspect the relation to open-set learning and recognition [1-3]: while DAUC provides a unified framework for understanding different types of uncertainty, including that caused by OOD examples, we would clarify that existing off-the-shelf algorithms [4] can also identify such uncertainty. What distinguishes DAUC is its ability to categorize uncertainty originating from various sources, such as ambiguous boundary examples, similar misclassified examples, and OOD examples, in a **unified framework**. Specialized algorithms could potentially further improve each element. We agree that this is an important area for future exploration and hope the reviewer concurs with our prioritized scope of the current work.
>
> We’ve added the discussion and relation on the open-set learning to our manuscript.
> ### Q4: Impact of validation data
> A4: The imbalance in misclassification will not affect the categorizations of DAUC, as DAUC leverages the confusion density matrix rather than individual elements of the matrix.
> The situation of *A certain class c1 … amount of misclassification* could be common in practice. For instance, in MNIST, there could be no examples from class 0 misclassified as class 7.
>
> DAUC’s categorizations are based on the sums over the off-diagonal and diagonal values of the confusion density matrix, respectively (refer to Eqn. (4) and (5)). Therefore, the **absence of specific misclassification instances does not directly impact DAUC's performance.** Additional empirical evidence is provided in **Figure 3 of the attached PDF file**.
> ### Q5: Scalability
> A5:
> While it is true that experimentation with more classes can be more computationally demanding, we wish to note that the scenarios the UQ methods primarily target often involve a limited number of high-stakes decisions (e.g., in healthcare, treat or not treat; in finance, buy, sell, or hold). In addition, the confusion density matrix calculation can be **highly parallelized**, such that it is always tractable based on modern computation resources even for more complex settings.
>
> To show the scalability explicitly, we **additionally experiment on the CIFAR-100 dataset**, with a special focus on the inverse direction. We use ResNet-18 to learn representations and identify the IDM and Bnd examples. We are then able to improve the prediction on those subsets. As we can observe in **Figure 1 of the attached PDF file**, DAUC scales effectively. By filtering similar examples in the training dataset, the prediction performance of the classifiers is enhanced.
>
> ---
> We hope our response can be helpful, and we are more than willing for any further clarification and discussion.
>
> ---
> *References*
>
> [1] *Zhang, Jingyang, et al. "OpenOOD v1. 5: Enhanced Benchmark for Out-of-Distribution Detection."*
>
> [2] *Yang, Jingkang, et al. "Openood: Benchmarking generalized out-of-distribution detection."*
>
> [3] *Yang, Jingkang, et al. "Generalized out-of-distribution detection: A survey."*
>
> [4] *Djurisic, Andrija, et al. "Extremely simple activation shaping for out-of-distribution detection."*

---

> ### Author Response · Authors · 2023-08-17
> **Dear Reviewer 7E4k**
>
> We truly appreciate the time and effort you have devoted to reviewing our paper. Following the feedback you and other reviewers provided, we've worked diligently to address the raised concerns. As we are approaching the close of the discussion phase, could you please let us know if there are any remaining questions or concerns? We are keen to address any outstanding questions with our utmost effort should there be a need.
>
> Sincerely,
>
> Authors

---

> > ### Comment · Reviewer_7E4k · 2023-08-18
> > **Post-rebuttal**
> >
> > Thank you for your detailed responses. All my questions have been addressed well. I will be increasing my score from 5 to 6.

---

> > > ### Author Response · Authors · 2023-08-19
> > > **Dear Reviewer 7E4k**
> > >
> > > We sincerely appreciate your encouraging feedback.
> > > Once again, we thank you for your time, effort, and consideration in reviewing and improving our paper.

---

### Official Review · Reviewer_hHZH · 2023-07-05

**Soundness:** 3 good
**Presentation:** 3 good
**Contribution:** 2 fair
**Rating:** 6
**Confidence:** 4

**Summary:**

I have read the other reviews and all rebuttals. The other reviewers are positive overall, and the authors provided very detailed and thorough rebuttals. I have increased my score from "5: Borderline accept" to "6: Weak Accept".
***
***
***
***


The authors propose an approach for categorizing examples which are flagged as uncertain by uncertainty estimation methods (e.g., deep ensembles, MC-dropout) into three classes: out-of-distribution examples (OOD), boundary examples (Bnd) and examples in regions of high in-distribution misclassification (IDM). They propose  a method for post-hoc uncertainty categorization, utilizing latent space / feature-space based density models.

**Strengths:**

The paper is very well written overall, I enjoyed reading it. More or less everything is clearly described and explained. The authors definitely seem knowledgeable, they have done a thorough job putting together the paper. I liked the color coding in Section 4.

Section 3.3 is interesting. The proposed confusion density matrix is novel (I think) and makes some intuitive sense, it is a pretty neat idea.

The example in Section 4.1 is quite neat, it provides good intuition.

The use case in Section 4.4 seems potentially useful.

**Weaknesses:**

I find it quite difficult to judge the technical contribution and impact of the proposed approach. Doing OOD detection based on feature-space density is not new (see e.g. Sun et al., "Out-of-distribution detection with deep nearest neighbors", ICML 2022), and I am not sure how useful the Bnd/IDM categorization actually is in practice.

The authors show two use cases in Section 4.3 and 4.4. However, in Section 4.3 the only concrete conclusion seems to be that DE is most apt at flagging the OOD class, but this is not really a new finding? And while the results in Section 4.4 do seem quite promising, the method is here only applied to MNIST?

In fact, the proposed method is never applied to any dataset with more than ~10 classes? Can it be difficult to apply to a dataset with e.g. 1000 classes? What happens to the confusion density matrix computation, does it become expensive? And what happens if the corpus C_{c1 --> c2} is empty for some pairs of classes?


Small thing:
- I found Section 3.3.1 a bit confusing at first, I had to get through Definition 4 before I understood the paragraph above. In particular, I was a bit confused by "The first situation where high uncertainty arises, is when g(x) is close to latent representations of validation examples that have been correctly assigned a label that differs from the predicted one".

**Questions:**

1. Can the proposed method be applied to datasets with a lot more classes (e.g. 1000 classes)? Does the confusion density matrix become expensive to compute? What happens if the corpus C_{c1 --> c2} is empty for some pairs of classes?

2. Does the validation set have be large, to ensure that the corpus C_{c1 --> c2} contains at least a few examples for each pair of classes? Does it have to be much bigger for e.g. 1000 classes than 10 classes?

3. For the use case in Section 4.3, could you clarify how the Bnd/IDM categorization is useful? What type of conclusions does it enable? Could it be used as evidence that uncertainty estimation method X is better than method Y?

4. Would it be possible to repeat the experiment in Section 4.4 on a larger dataset with more classes?


Minor things:
- Figure 1 caption: "into four classes" --> "into three classes"?
- Line 104: "for what a" --> "for why a"?
- Line 197: "an different" --> "a different".
- In Section 3.1 the term "inference-time"/"inference time" is used, but in the rest of the paper "test-time"/"test time" is used?
- In Definition 3 - 5 the full term "confusion density matrix" is used, but in the text in Section 3.3.1 and 3.3.2 just "confusion matrix" is used. Would it be more clear to always use the full term?

**Limitations:**

Yes.

---

> ### Author Rebuttal · Authors · 2023-08-07
>
> Thank you for your encouraging comments and appreciation of our paper's clarity, novelty, and presentation. We now address each specific question in turn:
>
> ---
> ### Q1: Working with more classes, computation, scalability, and additional experiments for Sec. 4.4.
> A1: Technically, there's **no strict computational barrier** preventing our method's extension to tasks with 1000 classes. While extending our method in those tasks would intensify computational demands, we would like to note that such a demand is tractable with modern computation resources, especially with our parallelized implementation: this is because the calculations of different elements in the confusion density matrix can be highly parallelized. On the other hand, the demand for interpretability often associates with high-stake tasks that typically present fewer classes, as evidenced in healthcare (e.g., treat or not treat) and finance decisions (e.g., buy, sell, hold).
>
> To further address the scalability concerns, we conducted **additional experiments** on the CIFAR-100 dataset. Given the limited time and the non-trivial efforts in creating boundary examples with corresponding gold labels, we would focus on the inverse direction.
>
> The results are shown in Figure 1 in our attached 1-page pdf response. First, we use ResNet-18 to learn representation and identify the suspicious IDM and Bnd examples. We are then able to improve the prediction on those subsets. We observe that DAUC scales well on the CIFAR-100 dataset — prediction performance of the classifiers is improved through filtering similar examples in the training dataset.
> ### Q2: What happens if the C_{c1 --> c2} is empty for some c1 c2?
> A2: In our work, C_{c1 --> c2} being empty means no validation examples from class c1 have been mistaken for class c2. This will lead to a zero value in the confusion density matrix. However, such zero values are not an issue and could be regularly observed. For instance, in the MNIST dataset, while class 1 examples might be misclassified as class 7, none from class 0 are mislabeled as class 7. In fact, we wish to note that DAUC's design ensures that its performance will not be critically affected by the absence of specific misclassification types. This is because the **categorizations of DAUC are based on the sums over the off-diagonal and diagonal values of the confusion density matrix** respectively (cf. Eqn. 4 and 5).
> ### Q3: Does the validation set have to be large? Sensitivity to the number of classes.
> A3: As explained in the last response, zero values in the confusion density matrix can be common in practice. We would not impose additional conditions on the validation set more than the conventional requirements: the validation set should be representative of the test set.
>
> To further verify the intuition, we present **additional experiments** to evaluate the effectiveness of DAUC across various settings. With the results presented in Figure 2-3 of the attached PDF file, several things can be deduced:
>
> - Figure 2: Increase the **number of classes** in the Dirty-MNIST dataset
>   1. The performance of DAUC in OOD identification remains consistently high across varying numbers of classes. This demonstrates that DAUC is resilient in identifying OOD examples, even when the in-distribution examples span fewer classes.
>   2. DAUC's ability to identify Bnd examples also shows good performance across different class counts. The F1-score indicates an upward trend as the number of classes increases, illustrating DAUC's adaptability in complex scenarios.
>   3. The accuracy of IDM identification appears to be more dependent on the number of classes, with a decrease in performance when the class count is reduced. This sensitivity can be explained by the accompanying decrease in the validation set size.
> - Figure 3: changing the **training-validation split** in the Dirty-MNIST dataset.
>
>   We varied the proportions of the training set, allocating between 50% to 90% of the data for training, with the remaining designated for validation. Across these configurations, DAUC consistently demonstrated high performance, underscoring its resilience to changes in the training-validation distribution. Particularly noteworthy is DAUC's ability to reliably identify both Bnd and OOD examples, regardless of the data split. However, as the validation set size decreased, we observe a decline in the accuracy of IDM identification. This trend mirrors our earlier observations where the performance dipped as the number of classes decreased.
> ### Q4: How is the Bnd/IDM categorization useful? Compare UQ methods.
>
> A4: In Section 4.3, we explored the types of examples that various uncertainty methods identify as suspicious, focusing on interpretation rather than ranking different methods. Our findings provide insights for further enhancements:
> - OOD Category: If a prediction is uncertain and falls into the OOD category, gathering more training data and labels can enhance predictive performance on similar examples.
> - Bnd Category: Uncertain predictions in the Bnd category signal that minor changes in the latent representation may alter the prediction. This characteristic is important for high-stakes scenarios like healthcare where rapid changes in decisions can occur.
> - IDM Category: This reveals that the current model struggles with accurate predictions for specific examples. In this case, collaboration with other models or expert assessments are practical ways to improve performance.
>
> While we demonstrated DAUC’s ability using the DE, MCD and BNNs methods, we wish to note that our design is generally compatible with any UQ method. As a model-agnostic approach, **DAUC paves the way for selecting the appropriate UQ method according to user specifications and preferences.**
>
> ---
> We hope these clarifications address your concerns and further illuminate our ideas. Should there be any additional questions or concerns, we are more than willing to provide further explanations.

---

> > ### Comment · Reviewer_hHZH · 2023-08-14
> >
> > Thank you for this very detailed response.
> >
> > I have read the other reviews and all rebuttals. The other reviewers are positive overall, and the authors provided very detailed and thorough rebuttals.
> >
> > I will increase my score from "5: Borderline accept" to "6: Weak Accept".

---

> > > ### Author Response · Authors · 2023-08-14
> > > **Reply by Authors**
> > >
> > > We sincerely appreciate your encouraging feedback. In our updated manuscripts, we would explicitly include the previously discussed questions to further enhance clarity.
> > >
> > > Thank you once again for your time, effort, and consideration in reviewing and improving our paper.

---

### Official Review · Reviewer_sVgx · 2023-07-06

**Soundness:** 2 fair
**Presentation:** 3 good
**Contribution:** 3 good
**Rating:** 6
**Confidence:** 4

**Summary:**

Paper introduces a framework to detect and categorize different model uncertainty types in classification setting. Proposer model-agnostic (with some assumptions of model structure) uncertainty quantification (UQ) relies on kernel density estimation on latent space representation by defining scores for OOD, Bnd, and IDM examples. Proposed approach is empirically evaluated using toy data as well as real MNIST and UCI datasets. Some of the limitations of existing UQ are examined as well as ability to use new uncertainty categorisation as flagging method to filter out training example to learn more accurate models.

**Strengths:**

To my knowledge, paper presents a novel approach to categorise different predictive uncertainty types in classification setting. The proposal is well-structured with appropriate theoretical background, and especially practical evaluation which includes both toy illustration of properties as well as real data evaluation of MNIST variants and UCI datasets. Proposed framework shows simple yet effective solution that seems to improve existing approaches in practice and is bringing new knowledge for analysing existing UQ methods.

Summary of strengths
- Novel view on uncertainty quantification
- Work in practice, at least in given classification tasks
- Well-made empirical evaluation (see also weaknesses)

**Weaknesses:**

There are some weaknesses that could further improve the presentation and usefulness of proposed framework. Analysis of different model architectures (i.e. latent space) with proposed density estimation could be further extend to show how these behave with different level of uncertainty in the datasets to justify the modelling choices. Also, the analysis of verifying DAUC concentrates only on MNIST variants, but it could be useful to examine different types of data with different noise and uncertainty levels.

Summary of weaknesses
- Limited analysis of model assumptions and density estimation
- Somewhat limited type of datasets and model architectures evaluated
- Some polishing of text here and there (see questions)


**Questions:**

- Table 1: Could you ellaborate why other related approaches could not be used to improve the prediction? (if threshold is determined for the uncertainty estimates these methods are utilising). I see prediction improvement more related to use case than general property of particular method directly. It might be not give better predictive accuracy than proposed method, but is doable.
- Figure 4: What is clustering/projecting method used? Maybe mention in the text.
- Figure 5 & 6: How error bars are calculated? Maybe mention in the text.

Minor
- DAUC is developed for classification tasks in specific. Maybe that could be mentioned in the abstract.


**Limitations:**

Some of the assumptions and limitations of proposed framework are listed in the supplementary material in relation to model structure and latent space density estimation. Furthermore, the usefulness of UQ are broader sense are mention as an important part of safety critical AI applications. No negative impact identified.

---

> ### Author Rebuttal · Authors · 2023-08-07
>
> We sincerely appreciate the reviewer's thoughtful insights and constructive feedback. We will respond to each of the questions in turn.
>
> ---
> ### Q1: Dataset and model architecture choices.
> A1:
> Our primary directive in selecting the Dirty-MNIST dataset was to anchor our work in a realm of reproducibility. Dirty-MNIST, beyond its public accessibility, is uniquely positioned with gold labels for boundary classes and OOD examples. This distinction positions it as an ideal choice in evaluating DAUC.
>
> In regard to our model architecture choices, we use a prevailing structure referring to the implementation of MCD, BNNs, and DE — the uncertainty quantification methods we benchmarked in our paper. We would like to emphasize, as highlighted in Table 1, that DAUC is designed to be **flexible in terms of model structure**. It functions effectively under conditions where Assumption 1 can be met, a scenario commonly satisfied when using prevailing neural network architectures.
>
> **[Supplementary Experiments in Appendix]** Recognizing the importance of demonstrating the broader applicability of DAUC, we enriched our evaluation with the Dirty-CIFAR dataset. Limited by the space, we postponed details of this additional evaluation in Appendix C.4 for your reference. We would also make this dataset publicly available.
>
> **[Extended Analysis in the Attached PDF]** In order to verify the scalability of DAUC, we conduct **additional experiments on the CIFAR-100 dataset** where the number of classes scales up to 100. Given the limited time and the non-trivial efforts in creating boundary examples and corresponding gold labels, we would focus on the inverse direction as a downstream application in demonstrating DAUC’s scalability.
>
> The results are shown in **Figure 1 in the attached 1-page pdf response**. For those experiments, we use ResNet-18 to learn representation and identify the IDM and Bnd examples marked as suspicious, and then improve the prediction on those subsets. DAUC scales well on the CIFAR-100 dataset — prediction performance of the classifiers can be improved through filtering similar examples in the training dataset.
>
> ### Q2: Why other related approaches could not be used to improve the prediction?
> A2:
> We would like to note that **DAUC’s capability of improving prediction is benefited by the uncertainty categorization**: To make this clear, we could consider a counter-example of why may other methods could fail in improving prediction: some methods may prioritize the predictions on the OOD examples as the most uncertain. E.g., the DE method tends to flag OOD examples as the most uncertain (as depicted in Figure 5).
>
> In order to improve the predictions, we could use the categorization of uncertainty provided by DAUC: Since the original classification model is trained with the entire dataset and minimizes overall prediction error, it may perform poorly on the minor examples. And DAUC is able to identify the source of uncertainty in such cases. We use this insight to further improve the prediction of those examples with a specialized focus on them.
>
> ### Q3: Figure 4: What is the clustering/projecting method used?
> A3:
> We used the t-SNE clustering method in generating the low-dim projection for the ease of visualization in Figure 4. We have updated the manuscript accordingly to make it clear.
>
> ### Q4: How error bars are calculated?
> A4:
> We used 10 repeated runs in our experiments and reported the mean and standard deviation in tables and figures. We have updated the captions to explicitly explain how error bars are generated in the manuscript.
>
> ### Q5: DAUC is developed for classification tasks in specific. Maybe that could be mentioned in the abstract.
> A5:
> It is true that DAUC is designed for uncertainty categorization in classification tasks. We have updated the abstract and introduction to make it clearer.
>
> ---
> We hope that these clarifications address your concerns, and we are happy to have further discussions would they remain unclear.

---

> > ### Comment · Reviewer_sVgx · 2023-08-16
> > **Rebuttal response**
> >
> > I have checked the rebuttal and other reviews. Authors have clarified most of my concerns and I am positive overall. To clarify Q2/A2 in
> > relation to Table 1, I meant that in general sense, other related methods could also be able to improve the prediction by flagging uncertain
> > examples, but as stated in the manuscript categorised uncertainty can provide more rich and robust ways to do it. Maybe you could consider to change the column title from "Improve prediction" to "Robust flagging to improve prediction" or something similar.

---

> > > ### Author Response · Authors · 2023-08-16
> > > **Response by Authors**
> > >
> > > We sincerely appreciate the reviewer's follow-up response and suggestion.
> > >
> > > We agree it is important to highlight that the advantage of improving prediction performance on subclasses of examples stems from our proposed method DAUC, which provides a fine-grained uncertainty categorization. In our revision, we would title this column "Fine-grained Prediction Improvement", to enhance clarity.
> > >
> > > Once again, thank you for your insightful comments and suggestions for refining our manuscript. Should there be any remaining concerns, please kindly let us know and we would do our utmost to address them.

---

### Author Rebuttal · Authors · 2023-08-07

We extend our sincere gratitude to all reviewers for their insightful comments, valuable suggestions, time, and efforts in evaluating and improving our paper.
We thank all reviewers for their affirmation of our work’s **novelty** (reviewers: sVgx, hHZH, 7E4k), **presentation** (reviewers: sVgx, hHZH, 7E4k, AvJq), **evaluation** (reviewers: sVgx, 7E4k), and **contribution** (reviewers: sVgx, 7E4k, AvJq).

----

To address the concerns raised by reviewers, we would respond to each of their questions respectively. Below, as a general response, we aim to outline the **key revisions and additional experimentation conducted by far**:

#### **Supplementary Experimental Evaluation** [Please refer to the attached PDF]

1. To stress-test the scalability of DAUC, we carried out additional experiments using the CIFAR-100 dataset.
2. To highlight the efficacy of DAUC across different number of classes, we conducted incremental studies with the Dirty-MNIST dataset.
3. We demonstrated the robustness of DAUC through a study varying the sizes of the validation dataset.

#### **Revised Manuscript for Clarity** (Include but not limited to)

1. We updated our introduction, Figure 1, and method sections to illustrate the motivation of our three-class categorization. This motivation stems from distinct characteristics manifested by the confusion density matrix.
2. We additionally discussed the relationship between DAUC, OOD detection algorithms, and open-set problems in our related work section.
3. We added a section in our appendix that explicitly explains the workflow of DAUC to enhance clarity: DAUC categorizes uncertain examples flagged by plug-in UQ algorithms.
4. We updated our manuscript to make the definition of Model-Agnosticity in our context clearer: it is used to express the applicability of DAUC to diverse UQ methods.


----

We hope our clarification and additional empirical studies could address the concerns raised by reviewers. Should there be any leftover questions, please let us know and we will make every effort to address them during the subsequent discussion period.

---

> ### Author Response · Authors · 2023-08-15
> **Further Discussions and Feedback Welcome**
>
> Dear Reviewers,
>
> We deeply appreciate the insights you've shared during the review process. Following our revisions and previous responses, we are genuinely curious if we have adequately addressed the concerns you raised.
>
>
> Should there be any leftover questions, concerns, or areas you feel need more clarification, please do not hesitate to let us know. We greatly respect your insights and stand ready to make any additional refinements based on your feedback.
>
> Best regards,
>
> Authors

---

### Decision · Program_Chairs · 2023-09-21

**Decision:**

Accept (poster)

**Comment:**

All reviewers agree that this paper makes an interesting and substantial contribution to uncertainty quantification. A few issues have been raised in the reviews, but these could mostly be clarified in the rebuttal. During the review phase, the authors made quite some changes in the paper, which is not uncritical, because these changes cannot be verified (unlike for journal papers, there is no official revision or possibility of a re-review). That said, their effort to comply with the reviewers' comments and suggestions is of course appreciated. Overall, all reviewers are on the positive side and in favour of acceptance.